# Interpretive monitoring in the caudate nucleus

**Marianna Yanike[1]\*, Vincent P Ferrera[1,2]**

[1]Department of Neuroscience, Columbia University, New York, United States;
[2]Department of Psychiatry, Columbia University, New York, United States

**Abstract** In a dynamic environment an organism has to constantly adjust ongoing behavior to adapt to a given context. This process requires continuous monitoring of ongoing behavior to provide its meaningful interpretation. The caudate nucleus is known to have a role in behavioral monitoring, but the nature of these signals during dynamic behavior is still unclear. We recorded neuronal activity in the caudate nucleus in monkeys during categorization behavior that changed rapidly across contexts. We found that neuronal activity maintained representation of the identity and context of a recently categorized stimulus, as well as interpreted the behavioral meaningfulness of the maintained trace. The accuracy of this cognitive monitoring signal was highest for behavior for which subjects were prone to make errors. Thus, the caudate nucleus provides interpretive monitoring of ongoing behavior, which is necessary for contextually specific decisions to adapt to rapidly changing conditions.

## Introduction

We adapt to different situations by sorting information into behaviorally meaningful categories that can change rapidly across contexts. This categorization process allows us to interact with such environments. Eager to get to an important meeting, we drive fast, but the actual speed depends on the speed limits along the way. In this situation, categories are delineated by a category boundary (i.e., speed limit) that varies with different environments. Flexible categorization requires continuous monitoring of the changes in the environment to ensure that classification reflects current task demands, even after categories are well learned. How the brain monitors such complex cognitive behavior is not well understood. The basal ganglia, particularly the caudate nucleus, are known to be essential for flexible behavior (*Wise et al., 1996*; *Barnes et al., 2005*) and together with the interconnected prefrontal and parietal cortices play an important role in categorization (*Poldrack et al., 1999*; *Poldrack et al., 2001*; *Seger and Cincotta, 2005*; *Freedman and Assad, 2006*; *Seger, 2008*; *Antzoulatos and Miller, 2011*; *Ashby and Maddox, 2011*; *Mendez et al., 2011*; *Merchant et al., 2011*). Many psychiatric and neurological disorders that compromise the caudate nucleus are characterized by impairment in cognitive flexibility (*Knowlton et al., 1996*; *Shohamy et al., 2004*; *Montoya et al., 2006*). Human and animal studies have provided extensive evidence for the critical role of this structure in category learning (*Seger and Miller, 2010*; *Antzoulatos and Miller, 2011*). Neurobiological models of categorization have suggested that the caudate nucleus, together with interconnected cortical and subcortical structures, contributes to the maintenance and switching of rules that guide categorization (*Maddox and Ashby, 2004*; *Ashby and Ennis, 2006*).

Despite the established role of the caudate nucleus in flexible behavior by linking actions and outcomes (*Hikosaka et al., 2000*; *Yin et al., 2005*; *Graybiel, 2008*; *Balleine and O'Doherty, 2010*), its role in the monitoring of such behavior is less clear. Recent studies have highlighted the strong contribution of the caudate to post-action evaluation by monitoring behavioral performance based on reward information (*Lau and Glimcher, 2007*; *Ding and Gold, 2010*; *Thorn et al., 2010*; *Kim et al., 2013*). The general idea is that the caudate detects a mismatch between expected and actual outcomes

\*For correspondence: marianna@vis.caltech.edu

**Competing interests:** The authors declare that no competing interests exist.

**Reviewing editor**: Ranulfo Romo, Universidad Nacional Autonoma de Mexico, Mexico

**eLife digest** The ability to adapt behavior in a changing environment is a hallmark of intelligent systems. From adjusting our driving speed to match road conditions to responding to a last-minute change of plans, mental flexibility underpins much of our day-to-day functioning.

To perform optimally, an animal must continuously monitor its own behavior and adjust it according to circumstances. A region of the brain called the caudate nucleus is thought to contribute to this process by keeping track of the relation between an action and its outcomes, but it is not clear how it monitors cognitive aspects of ongoing behavior.

Yanike and Ferrera have clarified this process by recording electrical activity from the caudate nucleus in two monkeys as they categorized visual stimuli. The monkeys viewed a moving stimulus and classified it as 'fast' or 'slow' relative to a reference speed that varied from trial to trial. The monkeys were trained to use two different references speeds and were told which reference speed to use at the start of each trial. They used an eye movement to indicate their decision.

Most neurons within the caudate nucleus responded after the monkey had made a decision, suggesting that these neurons might be involved in evaluating the decision that had just been made. The response of the neurons depended on the stimulus speed, and also on the category (fast or slow) in which the stimulus belonged. This observation indicates that the caudate nucleus tracked the context (reference speed) as well as the stimulus speed.

Yanike and Ferrera also showed that the response of the entire population of caudate neurons could be decoded to reveal both the speed of the stimulus and whether the monkey had categorized it as fast or slow. This shows that after a decision has been made, neurons continue to signal both the stimulus and the context in which that stimulus was presented. Such 'post-decision' monitoring is important for anticipating the outcome of the decision. Overall the results suggest that the caudate nucleus helps animals to adapt their behavior to rapidly changing circumstances by supporting decision-making that takes context into account.

and these prediction error signals tend to alert an organism about the overall level of behavioral performance. Some studies have shown that post-action neuronal activity maintained memory traces of specific actions, possibly linking them to outcomes (*Lau and Glimcher, 2007*; *Kim et al., 2013*). Others have found that post-action activity represented outcomes independent of specific actions (*Ding and Gold, 2010*). Monitoring signals are sensitive to behavioral context change (*Hikosaka and Isoda, 2010*) and can also incorporate uncertainty estimates (*Badre, 2012*; *Kepecs and Mainen, 2012*) known to modulate caudate activity (*Ding and Gold, 2012*; *Yanike and Ferrera, 2014*). Thus, a systematic evaluation of caudate monitoring signals is important for understanding its role in behavioral flexibility, when changes in context and outcome are frequent.

To study what aspects of cognitive flexible behavior are monitored in the caudate nucleus we recorded neuronal activity during a categorization task in which decision criteria changed rapidly across trials. We evaluated post-decision signals both at the level of individual neurons and their population activity. We found that while individual neurons were highly context specific, their population activity across ensembles of neurons provided an accurate and separable read-out of sensory and cognitive aspects of ongoing behavior.

## Results

We trained two monkeys on a speed categorization task in which they categorized stimulus speed depending on the position of two different category boundaries (*Figure 1A*). Each boundary divided a set of random dot stimuli of 8 different speeds (2, 4, …, 16 deg/s) into 'fast' and 'slow' categories. On each trial, one of the boundary speeds (5 or 13 deg/s), selected randomly, was indicated by a visual cue at the beginning of the trial, prior to the motion stimulus. Therefore, depending on the visual cue, a particular speed of dot movement would switch from belonging to the fast or slow category. Two choice targets (red/green), the locations of which were randomized from trial to trial, were presented adjacent to the motion stimulus. Monkeys made saccades to the red target if they categorized the stimulus as 'slow' and to the green target if they categorized the stimulus as 'fast'.

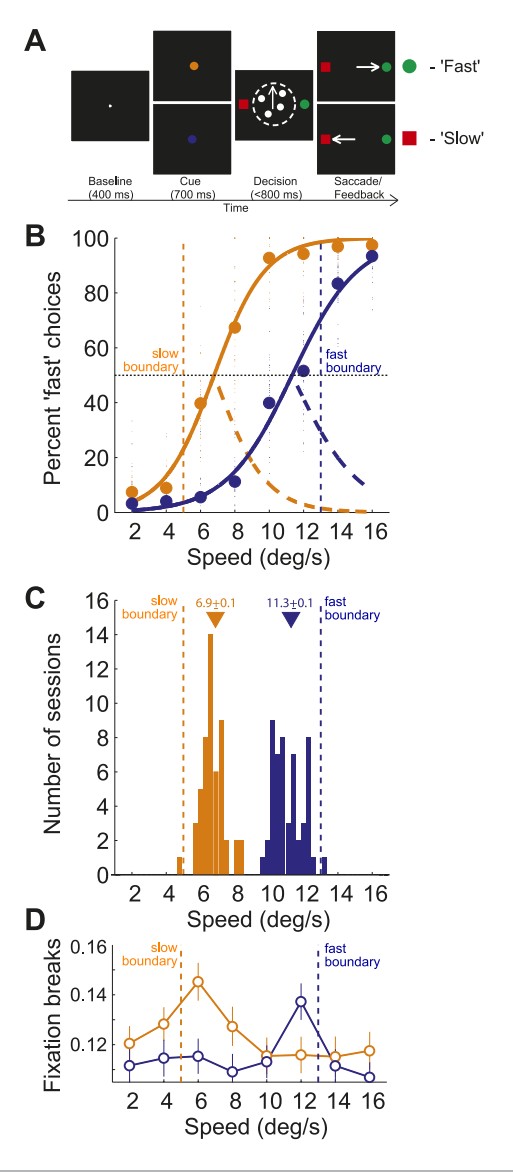

**Figure 1**. Behavioral task and performance. (**A**) Sequence of events in each trial of the speed categorization task. (**B**) Percentage of trials for which the stimulus speed was categorized as 'fast' for slow and fast boundary positions (orange/blue, respectively). Small circles represent choices from each session and large circles represent their average. Psychometric curves (solid lines) represent fits of Naka-Rushton functions together with the corresponding error rate (orange/blue dashed lines). (**C**) Histograms of PSEs from the psychometric curves across sessions sorted by the boundary position. PSE (i.e., the point of subjective equality) corresponds to the stimulus speed for which the animal was equally likely to classify the stimulus as 'fast' or 'slow'. Triangles represent the mean PSEs corresponding to internal estimates for each boundary position. (**D**) Proportion of fixation breaks of the total number of trials (open circles, mean ± SEM) for each speed sorted by the boundary position.

Both monkeys flexibly adjusted their categorization behavior depending on the position of the category boundary. We sorted behavioral data across animals and trials by the boundary position. We obtained two shifted psychometric functions (*Figure 1B*), suggesting that animals classified the stimuli using two different internal estimates of the boundary position. To estimate the internal representation of each boundary we found the stimulus speed for which the animal was equally likely to classify the stimulus as 'fast' or 'slow' (i.e., the point of subjective equality (PSE), *Figure 1B*). The per-session PSE's formed two non-overlapping distributions (*Figure 1C*) with means (slow: 6.9 ± 0.1; fast: 11.3 ± 0.12) shifted inward relative to the actual boundary speeds (5 and 13, respectively; *Figure 1C*). This implies that the monkeys' categorical decisions were less accurate for stimulus speeds near the boundaries, possibly due to uncertainty about the boundary position or its internal estimate. Consistently, the greater categorization uncertainty at the boundaries, expressed as an error rate, correlated with a higher rate of trials that the animal aborted without making a choice (broke fixation) (*Figure 1D*). Both the error rate and the rate of fixation breaks peaked for speeds 6 and 12, suggesting that subjects had greater difficulty categorizing these near boundary stimuli.

## Individual neurons monitored specific context

We analyzed a total of 155 presumed projection neurons from the associative caudate nucleus (*Figure 2A*) in two monkeys performing the speed categorization task (Monkey C: n = 91; Monkey F: n = 64). For each neuron only trials in the neuron's response field were included. The data from the two monkeys were combined as they were qualitatively similar. We found that the majority of caudate neurons were responsive after, and not prior to, the animal making a decision. Therefore we focused on two post-decision periods of the task, one right after the decision ('post-saccade', 0–400 ms after saccade onset, median saccade onset 309 ms) and another when the correctness of the decision was revealed at 800 ms after the decision onset ('reward', 0–600 ms after reward onset). We found that many neurons (71 out of 155, 46%, *Figure 2B*, top) showed significantly different activity between the fixation period and at least one of the post-decision periods of the task (bootstrap test, p < 0.01). Next, we asked whether caudate neurons represented category-related information regardless of the properties of the visual stimuli or their stimulus selectivity.

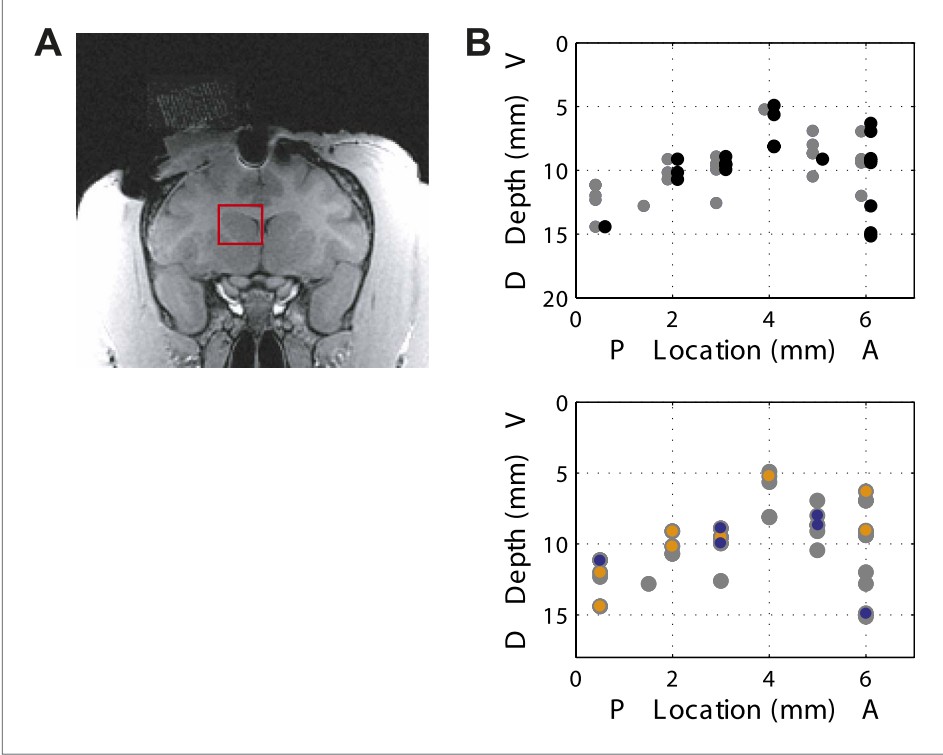

**Figure 2**. Location of recording sites. (**A**) Recording chamber showing access to the associative striatum (indicated in red square) in Monkey F. (**B**) (top) Distribution of recording sites for post-saccade (gray) and reward (black) neurons across two animals. AP = 0 corresponds to the anterior commissure. (bottom) Location of category-related neurons with selective response to slow (orange) and fast (blue) boundaries superimposed on the location of responsive neurons (gray).

We identified 38 out of 71 neurons that were sensitive to the category context (i.e., boundary position) during either one or both post-decision periods (*Figure 2B*, bottom). The firing rates of these neurons were significantly different between the two boundary positions for at least one speed either during the post-saccade ('psacc', n = 31) and/or reward ('rwd', n = 23) periods of the task (*Figure 3*, boostrap test, p < 0.05 Bonferroni corrected). *Figure 4* shows example neurons with activity during the post-saccade (A) and reward (B) periods with a significantly different average response to one out of 8 stimuli, corresponding to speed 12 (see *Figure 4—figure supplement 1* for other examples), for which they responded with either significantly higher (*Figure 4A*) or lower (*Figure 4B*) firing rate on trials with the fast compared to slow boundary positions (bootstrap test, p < 0.0001). These neurons responded similarly to most other stimuli, irrespective of the boundary position (*Figure 4A,B*, bottom). The majority of these post-decision neurons (psacc: 19/31, 61%; rwd: 15/23, 65%) discriminated significantly only one out of 8 stimuli and many of them (psacc: 22/31, 71%; rwd: 11/23, 48%) had significantly different neuronal activity for stimuli near the category boundaries, speeds 6 and 12 (*Figure 4E,F*, bottom).

To quantify the sensitivity to the boundary position, we calculated a category index for each neuron (CI; see 'Materials and methods') separately for each period of the task. The CI measures how well neuronal responses discriminate the same stimuli when presented with different boundaries. The index ranges between 0 and 1, with values close to 0 indicating weak discrimination and values close to 1 indicating strong differentiation between the two categories. Similarly to the example neurons, the CIs between the stimuli close to the boundaries (speeds 6 and 12) were different for most individual neurons in both periods of the task (*Figure 4C,D*). When averaged across neurons, the CIs for the two most difficult stimuli were the highest during both periods of the task (*Figure 4E,F*, top). The ability of individual neurons to multiplex their function according to the relevant boundary position is inconsistent with the known sensitivity of caudate neurons to the rate of reward coding, which varied similarly

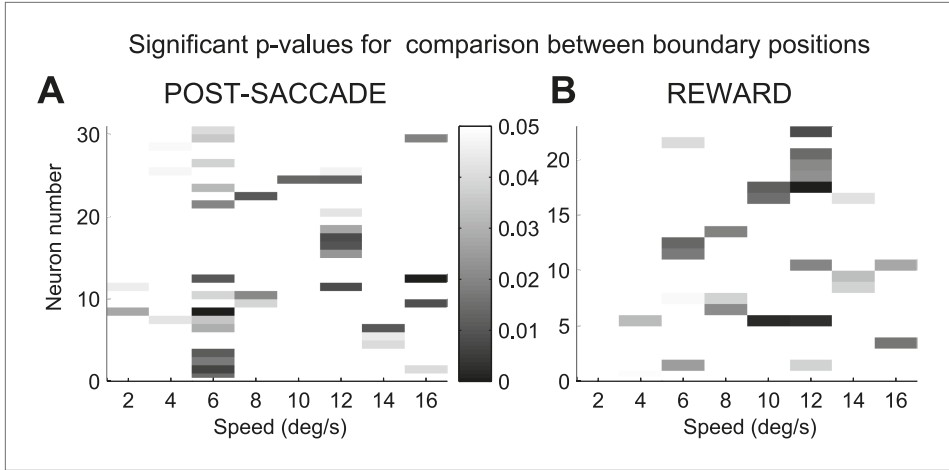

**Figure 3**. Sensitivity to the boundary position in caudate neurons. (**A** and **B**) Distribution of significant p-values indicating difference in spike count between slow and fast boundary positions for neurons with activity during the post-saccade (**A**) and reward (**B**) periods of the task. Only values of less than 0.05 are shown.

with two boundaries (***Cromwell and Schultz, 2003***; ***Hassani et al., 2001***). These results show that individual neurons were limited in their ability to monitor behavior across multiple boundaries. However, as a population, the neuronal activity of these neurons could differentiate stimuli near both category boundaries.

## The neuronal population read-out provided a global monitoring signal

Next we tested whether combined signals across pools of individual neurons could provide a reliable population code (***Pouget et al., 2003***) to monitor behaviors across multiple boundaries. We predicted on each trial the stimulus speed and category boundary position (for a total of 16 conditions: 8 × 2) based on the neuronal activity of populations of independently recorded neurons (see 'Materials and methods'). To evaluate the prediction accuracy, we used the proportion of correct estimates for either speed or boundary position. We found that on each trial, the population activity of ensembles of caudate neurons provided a reliable read-out of the two signals: the identity (speed) of the previously categorized stimulus and the context (boundary position) in which the stimulus was categorized with above chance accuracy during the post-saccade (***Figure 5A***, B; n = 31) and reward (***Figure 5C,D***; n = 23) periods of the task. To account for variable levels of the subjects' performance, we obtained the prediction accuracy for the speed and boundary position separately for correct only trials and for all trials (correct and incorrect) separately. On average, the prediction accuracies for the two signals were significantly better on correct trials compared to all trials for the post-saccade (***Figure 5A,B***; 1-way ANOVA (trial type), p < 0.0001) and reward (***Figure 5C,D***; p < 0.0001) populations. We also evaluated decoding accuracy as a function of the number of neurons (***Figure 5—figure supplement 1***). The decoding performance converged towards an asymptote faster for the boundary position (~5 neurons) compared to the speed (~10–15 neurons) in each neuronal population. This suggests a more redundant neuronal code for the boundary position.

To further quantify the relationship between the accuracy of the population read-out and subjects' behavioral performance for each stimulus, we determined the prediction accuracy of the stimulus speed by splitting trials into slow and fast boundary positions. We found that the accuracy of the population read-out for the identity of the previously categorized stimuli varied with the categorization difficulty at each boundary. This was shown by a significant interaction between the speed and boundary position (***Figure 6A,C***; 2-way ANOVA (speed, boundary), p < 0.001) in each neuronal population. To correlate subjects' categorization performance and the reliability of the population representation, we averaged the prediction accuracy between the slow and fast boundary positions across all speeds and plotted it as a function of the distance to the boundary for post-saccade (***Figure 6B***) and reward (***Figure 6D***) populations. The average prediction accuracy was the highest for the near boundary stimuli on the right, coinciding with the highest behavioral error rate (***Figure 1B,D***), then

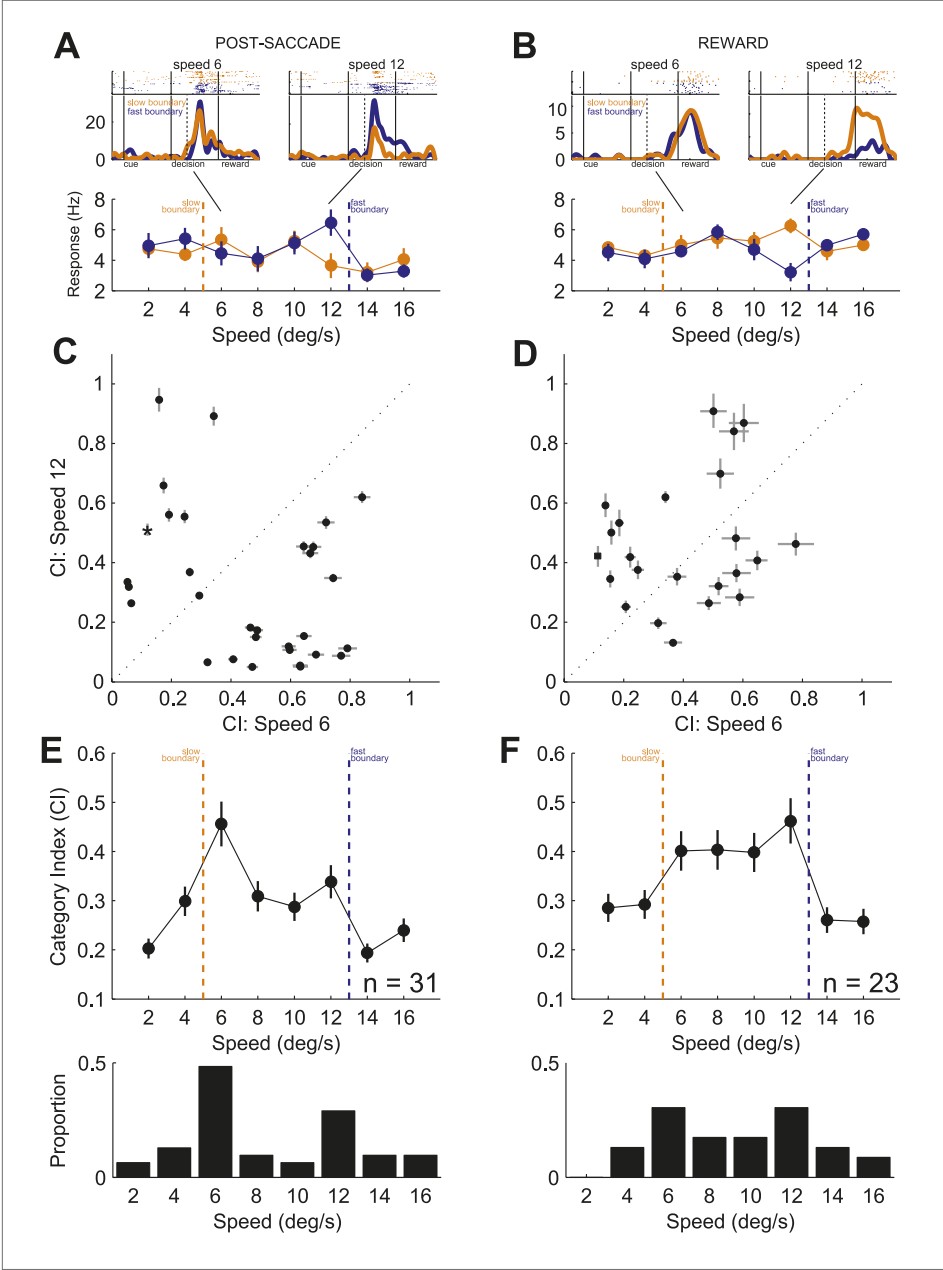

**Figure 4**. Representation of category signals. (**A**) Example neurons with activity during post-saccade (**A**) and reward (**B**) periods. (top) Spike raster plots, each row is one trial and each dot is detected spike, and spike density functions (mean ± SEM) in response to speeds 6 (left) and 12 (right). Black lines show periods of the task. Dashed black lines indicate average reaction time to saccade. (bottom) Average neuronal activity across stimuli sorted by the boundary position. Orange/blue dashed lines, actual boundary positions. Bars, SEM. (**C** and **D**) Scatter plots of CIs for speed 6 vs speed 12 across neurons with activity during post-saccade (**C**, n = 31) and reward (**D**, n = 23) periods. Example neurons in (**A** and **B**), star and square in (**C** and **D**), respectively. (**E** and **F**) (top) Average CI across speeds for neurons with activity during post-saccade (**E**) and reward (**F**) periods. (bottom) Proportion of neurons with a significant difference between spike counts across boundaries for each speed (bootstrap test, p < 0.05).

The following figure supplement is available for figure 4:

**Figure supplement 1**. Examples of two additional caudate neurons.

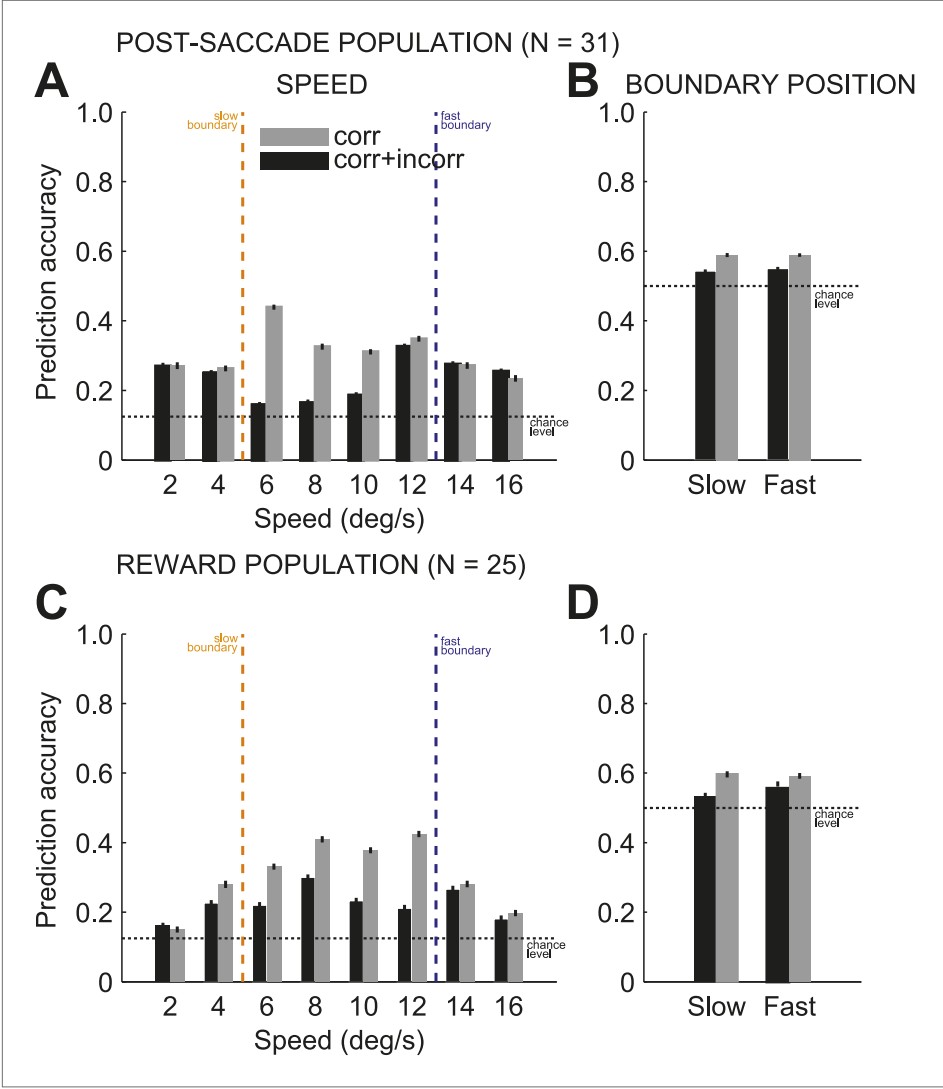

**Figure 5**. Prediction accuracy for speed and category boundary. The average prediction accuracy for speed (**A**, **C**), across two boundary position, and boundary position (**B**, **D**), across all speeds, for correct only trials (black) and all trials (correct and incorrect, gray) separately for the post-saccade (**A**, **B**) and reward (**C**, **D**) neuronal populations. Corresponding chance levels are shown in dashed line. Orange/blue dashed lines, actual slow/fast boundary positions.

The following figure supplement is available for figure 5:

**Figure supplement 1**. Prediction accuracy and population size.

decreased with the distance to the boundary. The population read out of the previous stimulus speed was strongly correlated with subjects' behavioral performance, expressed as an error rate, averaged across two boundaries for each speed (*Figure 1B*), similarly in both neuronal populations (*Figure 6B,D*; psacc: $R^2 = 0.85$; rwd: $R^2 = 0.76$). Thus, the population was best at maintaining the sensory representation of stimuli, for which the subjects were prone to make categorization errors.

We then compared the predication accuracy between trials when subjects categorized the same stimuli correctly and incorrectly for a subset of stimuli, speeds 6 and 12, with sufficient number of incorrect trials (see 'Materials and methods'). In both neuronal populations, the incorrect categorical judgments were followed by the failure to represent the identity of the preceding stimulus (*Figure 6A,C*). In contrast, correct categorization of the same stimuli, either near or far from the boundary, were followed by a reliable but different read-out of stimulus speed based on the population activity

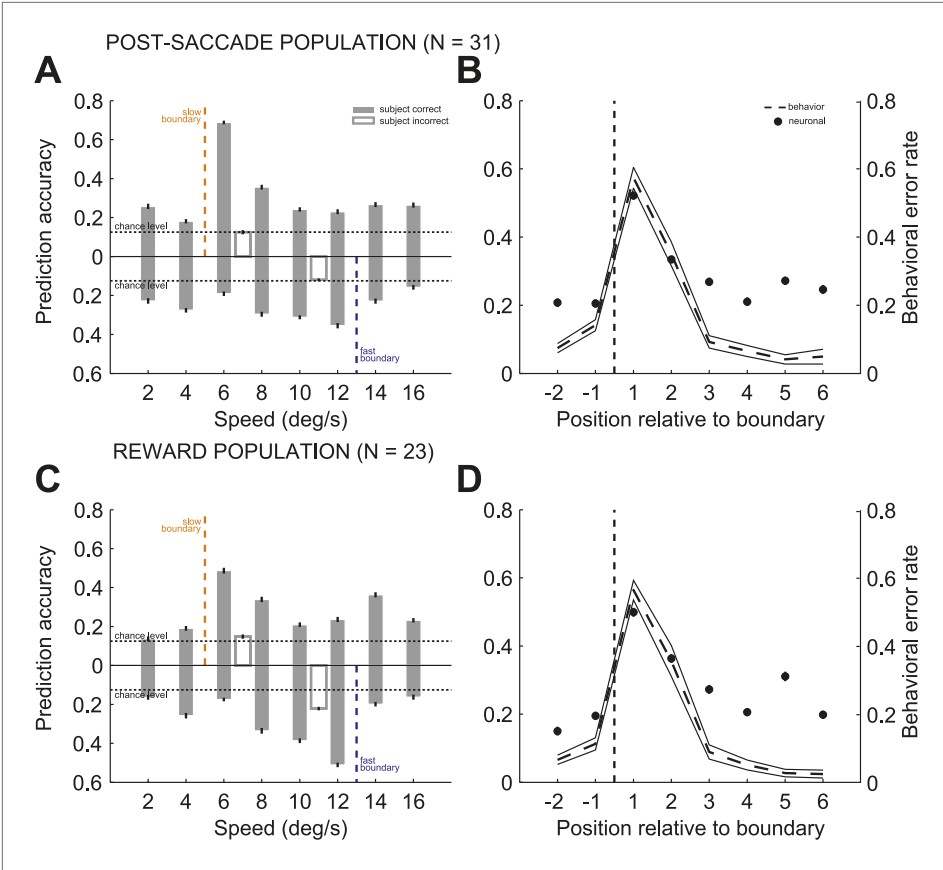

**Figure 6**. Population read out of speed and boundary position. (**A** and **C**) Proportion of correct estimates for each stimulus speed separately for slow (upper bars) and fast (lower bars) boundary positions for post-saccade (**A**) and reward (**C**) populations for correct only (gray) and incorrect (open) trials. (**B** and **D**) Average prediction accuracy for speed (black circles) and behavioral error rate (mean ± SEM, dashed and solid lines) as a function of stimulus' position to the boundary (dashed line) for each neuronal population. The prediction accuracy for the most extreme stimuli (speeds 2, 4, 14, and 16) deviated from the behavioral error rate function, possibly due to a greater perceptual uncertainty, about the identity of the stimulus or boundary cue, compared to the intermediate stimuli.

(2-way ANOVA [trial type, boundary], $p < 0.001$). These results suggest that outcome uncertainty near the boundaries can act as a gating mechanism to enhance representation of the boundary stimuli to potentially overcome decision uncertainty.

## The monitoring signal reflected behavioral relevance

We reasoned that if this neuronal code is cognitive and reflects categorization process, instead of reward processing, then even irrelevant information which could be pertinent for categorization should be processed in a task meaningful way. We asked if caudate neurons encoded category-irrelevant features of otherwise relevant stimuli and whether the post-decision evaluation of the irrelevant information reflected categorization difficulty. We tested this hypothesis by computing how well we could predict discrimination accuracy between two opposite directions of dot motions for each stimulus separately for the slow and fast boundaries based on neuronal activity (see 'Materials and methods'). We found that even those features of stimuli (i.e., dots direction) that were not linked to category identity were selectively suppressed when stimuli were near the boundary compared to when they were far from the boundary in both neuronal populations (see 'Materials and methods'; *Figure 7A,B*; 3-way ANOVA (direction, speed, boundary), $p < 0.01$). Similarly to single unit studies (*Hussar and Pasternak, 2009*), the post-decision population read-out in the caudate strongly reflected behavioral relevance of stimuli during categorization.

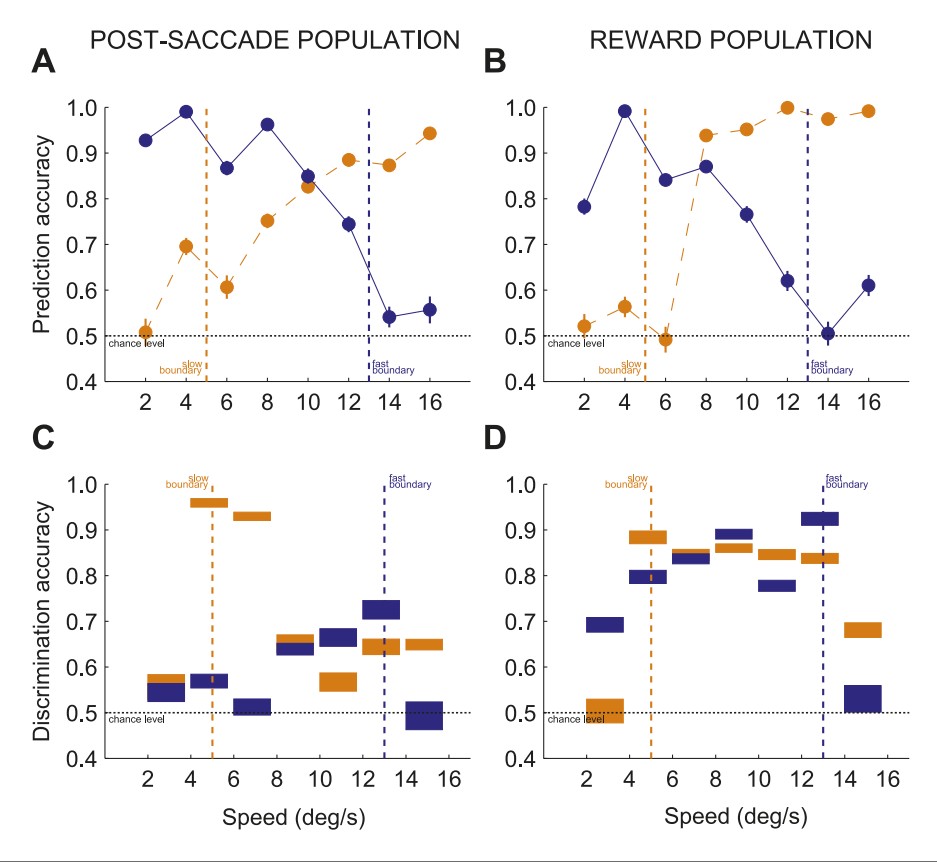

**Figure 7**. Contextual modulation during categorization. (**A** and **B**) Prediction accuracy for two opposite directions of dots motion (up or down) for each stimulus for post-saccade (**A**) and reward (**B**) neuronal populations. (**C** and **D**) Discrimination accuracy across pairs of neighboring speeds (seven pairs, 2–4, 4–6, …, 14–16) separately for trials with each boundary position for post-saccade (**C**) and reward (**D**) neuronal populations. The thickness of each bar corresponds to the average discrimination accuracy with ± SEM. Same notation as in above.

The following figure supplement is available for figure 7:

**Figure supplement 1**. Effect of categorical perception on discrimination of near boundary stimuli in individual neurons.

## The monitoring signal provided parsed representation of categorization context

The bias towards processing near boundary stimuli in the caudate population code suggests that it could represent the position of each boundary, either actual or inferred. The actual boundary positions (speeds 5/13 for slow/fast boundaries respectively) were never explicitly given and the subjects had to infer the positions of each boundary through training. We tested whether caudate population represented the position of each boundary by comparing the discrimination accuracy for the seven pairs of neighboring speeds (2–4, …, 14–16) separately for each boundary and neuronal population (see 'Materials and methods'). We reasoned that the greater discrimination accuracy between pairs of neighboring speeds would correspond to the more accurate neuronal representation of the internal estimates of the boundary position. We found that immediately after the decision, the post-saccade population activity provided signals specific to all behaviorally relevant boundary positions. *Figure 7C* shows that high discrimination accuracy occurred for stimuli across the actual boundary positions (speed pairs 4/6 and 12/14), and across the average internal estimates of the boundary positions (6/8 and 10/12, respectively). *Figure 7D* shows a different pattern during reward delivery. The strength of the discrimination accuracy peaked for the speeds across the actual boundary positions (4/6 and 12/14), but remained elevated for intermediate speeds regardless of the distance to the

boundary. These differences in population read-out codes possibly reflect different putative synaptic pooling mechanisms, with the pooling during the reward period better suited to linking stimuli with dual class membership to the received outcome. We confirmed that this signal reflects categorical perception as opposed to representation of stimulus identity. We reasoned that any two stimuli should be discriminated better when they are judged to be in a different class as opposed to being in the same class. This is often referred to as the boundary accentuation effect (*Goldstone, 1994*). The results in *Figure 7C,D* confirmed this prediction, the average discrimination accuracy between the near boundary speeds (4/6 and 12/14) being significantly greater when the stimuli were classified in two different (slow vs fast) compared to the same (slow or fast) categories in both populations (psacc: $0.86 \pm 0.02$ vs $0.63 \pm 0.02$; rwd: $0.9 \pm 0.01$ vs $0.82 \pm 0.01$, permutation test, $p < 0.001$). To understand the representation at the level of individual neurons, for each neuron we calculated a ratio difference index (Ratio Diff) measuring the difference in firing for two pairs of speeds (4/6 and 12/14) separately for slow and fast boundary trials. Consistent with the population findings, individual neurons had significantly larger differences in firing rate to the near boundary stimuli when in different categories compared to the same category both during the post-saccade (boostrap test, $p < 0.04$) and reward ($p < 0.01$) periods of the task (*Figure 7—figure supplement 1*). These results suggest that the caudate neuronal code parses information into behaviorally relevant categories and that the code is most informative for the stimuli with the most erroneous performance.

## Discussion

We found that caudate neurons monitored flexible categorization behavior by providing distinct representation of decision-pertinent variables, from stimulus specific features to more cognitive signals reflecting internal estimates of decision variables. The monitoring signal in the caudate was not only accurate at representing ongoing behavior, but it also interpreted behavioral consequences in light of changing demands of the task. The accuracy of the monitoring signal was inversely related to behavioral proficiency, suggesting that the caudate nucleus allocates more efficient coding for changing and/or uncertain information critical for behavioral flexibility. These findings provide direct evidence for the role of the caudate nucleus in online monitoring of complex cognitive behavior, possibly allowing for contextually specific decisions to adapt to rapidly changing context (*Daw et al., 2006*; *Hikosaka and Isoda, 2010*; *Pearson and Platt, 2013*).

It is well-established that the caudate nucleus supports cognitive function, including categorization, by utilizing reward or motivation information. However, how cognitive and reward signals interact to support complex behavior is still unclear. In this report, we showed that the monitoring signals in the caudate nucleus reflecting cognitive variables were shaped by reward modulation, as more accurate representation correlated with the most variable reward outcome. We found that individual neurons were highly selective at linking sensory and reward information. In other words, the neurons did not generalize across sensory inputs to code reward-related information. However, their population response provided a reliable and separable representation of both cognitive and motivational signals on a trial-by-trial basis. The reported monitoring signals in the caudate can reflect the outcome prediction and/or the rate of reward coding (*Hollerman et al., 1998*; *Hassani et al., 2001*; *Cromwell and Schultz, 2003*). The motivational signals potentially shape categorical representation in the caudate to monitor context specific decisions to adapt to rapidly changing conditions.

Our findings are consistent with the growing evidence for the strong contribution of the caudate nucleus to post-decision monitoring and evaluation (*Lau and Glimcher, 2007*; *Ding and Gold, 2010*; *Thorn et al., 2010*). Traditional views postulate that the striatum, which includes the caudate nucleus, contributes to behavioral evaluation through updates based on differences between the observed and expected outcomes through reinforcement learning (*Daw et al., 2006*; *Williams and Eskandar, 2006*). Recent views suggest that such updates can also incorporate uncertainty estimates, similar to the prefrontal cortex (*Badre, 2012*; *Kepecs et al., 2008*; *Kepecs and Mainen, 2012*). Caudate neuronal activity is sensitive to both stimulus (*Ding and Gold, 2012*) and outcome uncertainty (*Yanike and Ferrera, 2014*), which can potentially shape more efficient coding of changing or uncertain information.

Our findings that the caudate nucleus provides online monitoring of flexible behavior are consistent with a recent study showing that individual caudate neurons coded values of visual objects in a flexible manner (*Kim and Hikosaka, 2013*). Specifically, neurons in the head of the caudate nucleus represented changes in recent value of objects while monkeys made saccades to visual objects with different values and inactivation of this structure disrupted their behavioral preference for high value objects. While we

also recorded in the anterior part of the caudate nucleus (see 'Materials and methods'), our experimental paradigm was different. During the categorization task, monkeys had to maintained representation of both stable (i.e., always far from boundaries) and changing (i.e., near the boundaries) values of dot stimuli to perform successfully, well after the initial learning had occurred. Therefore, our findings are complementary to the aforementioned study because we showed that the caudate nucleus provides abstract cognitive monitoring signals beyond flexible reward associations. Taken together, these results suggest that the anterior caudate plays an important role in the change detection network (*Isoda and Hikosaka, 2011*; *Pearson and Platt, 2013*) contributing to a rapid adjustment of behavior.

Our findings that the representation of category-relevant information was enhanced while category-irrelevant information was suppressed near the boundaries are consistent with the possible role of the caudate nucleus in the working memory updating. Computational models (*O'Reilly and Frank, 2006*) have suggested that the striatum contributes to a selective gating of information flow into the working memory in the prefrontal cortex. The present findings suggest that such selective gating in the caudate nucleus can occur via a neuronal population code, which allows for independent read-out of all task variables by downstream structures. In our experimental paradigm the direction of the dots motion was irrelevant for the speed categorization and the animals were never explicitly required to utilize that information. Yet, the accuracy of the population read-out for the dots motion direction in the caudate was suppressed near the boundary compared to that far from the boundary. These results suggest that the caudate nucleus sorts information by its relevance to the task at hand, possibly automatically, to potentially affect ongoing behavior (*Badre, 2012*).

Finally, our findings suggest that unrevealing the nature of neuronal population code can greatly benefit our understanding of the neural basis of complex cognitive behavior. We found that individual neurons provided a context-specific code, while their population read-out covered all aspects of behavior and multiple separable signals could be reliably extracted. Previous studies on visual categorization have shown that representation of multiple categories in the prefrontal cortex at the level of single neurons was either distributed or sparse depending on how much the visual categories overlapped (*Cromer et al., 2010*; *Roy et al., 2010*). How these differences translate into population codes remains to be studied. The results of our experiment show that the caudate neuronal population code can reliably represent all aspects of cognitive complexity during flexible behavior.

## Materials and methods

### Surgical and recording procedures

Two adult male rhesus monkeys (*Macaca mulatta*, Monkey C: 8.2 kg and Monkey F: 11.5 kg) were used in the experiments. All 'Materials and methods' and treatments were in accordance with NIH guidelines and approved by the Institutional Animal Care and Use Committee at Columbia University and the New York State Psychiatric Institute. Prior to the experiments each animal was implanted with a scleral search coil, head post and recording chamber under aseptic conditions using isoflurane anesthesia. The animals received postoperative analgesics during postsurgical recovery. The positions of the recording chambers were guided by monkeys' individual MRI atlases. The recording chamber (20 mm in diameter) for Monkey C was placed on the scull over the arcuate sulcus positioned at stereotaxic coordinates 20 mm anterior and 15 mm lateral allowing access to the anterior caudate nucleus via the frontal eye fields (FEF). Monkey F was sequentially implanted with two different recording chambers. The first recording chamber (20 mm in diameter) was placed at 25 mm anterior and 18 mm lateral positioned over the acruate sulcus. The second recording chamber (20 × 30 mm oval) was centered at 15 mm anterior and 12 mm lateral. We used tungsten or glass coated electrodes with impedance ranging from 1–3.5 MΩ. The signals were amplified, filtered and passed to a real-time action potential detection. Action potentials were converted to TTL pulses that were stored together with the behavioral data. We also stored individual waveforms on each channel for further offline analysis. To identify the anterior caudate nucleus we used a number of criteria. We used depth measurements and identified the position of the caudate relative to the FEF. Also the dorsal edge of the caudate was identified by the presence of injury potentials. We identified the phasically active neurons by their low baseline activity (1–3 Hz). On each recording track we made sure to identify a tonically active neuron (4–8 Hz), in fact during some sessions we simultaneously recorded a pair of tonically and phasically active neurons. Post-saccade and reward neurons tended to be distributed without any distinct

spatial organization in the associative caudate (*Figure 2B*). Monkeys were trained to sit in a primate chair for the duration of the experiments with their heads restrained. They performed behavioral experiments and received liquid reward for correctly executing the behavioral task.

## Behavioral task

We trained two adult monkeys to categorize the speed of moving random dot patterns depending on the position of a category boundary (*Ferrera et al., 2009*). The stimulus set consisted of random-dot patterns moving at 8 different speeds (2, 4, 6, …, 16 deg/s) with coherence equal to 1. The direction of random dot motion also varied randomly. On each trial, dot direction was selected from a set of two opposite directions. Generally, the directions were 'up' and 'down' although other axes of motion were also tried. Animals were never asked to judge the direction of dot motion, thus we consider this stimulus dimension as category-irrelevant. On a given trial, monkeys judged the speed of motion as 'slow' or 'fast' depending on one of two reference speeds. Each trial started with a fixation cue presented at the center of the computer screen. After animals fixated for 400 ms (baseline period), one of two boundary cues (blue or orange squares) indicating the reference speed was presented for 800 ms (cue period), followed by the random-dot stimulus together with two spatially located targets (decision period). Monkeys were trained to associate speeds faster than the reference with the green target and slower than the reference with the red target. They indicated their judgment by making a saccade within 800 ms to one of the targets, the positions of which were randomized across trials. Feedback was provided at the end of the trial. Correctly categorized stimuli were followed by two drops of water and a high tone, while incorrectly categorized stimuli were followed by a low tone and no reward. The trials were separated by a 2000 ms inter-trial interval.

The task had a block-randomized design; each trial type was presented randomly from a block and animals had to complete each trial-type to progress to the next block. The full design comprised 64 trial types: 8 speeds × 2 directions × 2 boundaries × 2 target locations. On a small fraction of trials (~0.13) animals broke fixation during the decision period of the task without making a choice (fixation break trials). The fixation break trials were reshuffled with the different trials in a given block and were not immediately repeated. The average reaction time to abort on the fixation break trials was 348 ± 12 ms and was similar between the two animals (monkey: F, 328 ± 13 ms, and monkey C: 367 ± 18 ms; 1 tail *t* test p = 0.08). Trials on which animals broke fixation during the baseline or cue periods of the task were excluded from the analyses.

We used a memory guided saccade task with 8 spatial targets at 45° intervals to identify task-related neurons. We identified each neuron's response field by finding the spatial location which evoked the maximum firing rate during one of the task periods in the memory guided saccade task. In the speed categorization task, we placed one of the spatial targets in the response field and one in the location opposite to the response field of a neuron. Some of the cells that were responsive during the memory guided saccade task were not responsive during the categorization task (21/176 cells, 12%). These cells were excluded from the present analyses. We analyzed a total of 155 cells with the average of 494 ± 94 trials per session (range [285–1393]), which was similar between the two animals (Monkey F: 551 ± 92; Monkey C: 444 ± 105; 1 tail *t* test, p = 0.36). The monkeys performed on average similar number of trials with slow and fast boundary positions across all speeds (mean slow boundary: 226 ± 16 trials; mean fast boundary: 219 ± 15; paired *t* test p = 0.74) and on average 27 ± 2 trials for each speed. For each neuron, we only included trials in a neuron's response field during the categorization task.

## Data analysis

We identified all task related neurons by a bootstrap test (p < 0.05) comparing baseline firing rate during the initial fixation period (400 ms) with the average spike counts during each of four different task periods: cue (0–700 ms), decision (0 ≤ 800 ms), post-saccade (0–400 ms), and reward (0–600 ms). We focused out analyses on the post-decision activity and only included neurons with significant task-related activity either during 400 ms after saccade ('post-saccade' activity aligned to saccade onset) and 600 ms during reward ('reward' activity aligned to the reward onset) periods of the task. For each neuron, only trials in a neuron's response field were included. The data from the two monkeys were combined as they were qualitatively similar. The majority of these post-decision neurons (psacc: 19/31, 61%; rwd: 15/23, 65%) discriminated significantly only one out of eight stimuli and many of them had significantly different neuronal activity for stimuli on the inside near the category boundaries, speeds 6 and 12 (psacc: 22/31, 71%; rwd: 11/23, 48%).

## Single neuron metric

We calculated a category index (CI) as $|(R_{slow} - R_{fast})/(R_{slow} + R_{fast})|$, where $R_{slow}$ and $R_{fast}$ indicate the spike count to stimuli of different speeds (2, 4, …, 16) with slow and fast boundary position trials, respectively. This index ranges from 0 to 1, with values closer to 1 indicating a larger difference in activity for a given stimulus between slow and fast boundary positions. We used an unsigned category index, as we were focusing on the difference in activity, not the preference for either boundary position.

We calculated a ratio difference index (Ratio Diff) by taking the unsigned difference in the spike counts between two neighboring speeds separately for each boundary position. Because each neuron had stronger preference for one of two boundary positions, for each cell we selected the speed pair (4/6 or 12/14) with the largest difference and use that speed pair for the population analysis.

## Neuronal population metric

We sought to determine how accurately the information about the stimulus identity (speed) and relevant context (boundary position) were represented in neuronal activity across striatal neurons after animals made their categorical choices. We wanted to simultaneously decode the stimulus speed and boundary position from the population response to obtain an accurate estimate of each parameter on a single trial. The population response on a single trial could reflect a stimulus speed $s_j$ at a given boundary with $j$ = 1:16, where $j$ = 1:8, speeds with the slow boundary; and $j$ = 9:16, speeds with the fast boundary. We use a Poisson independent decoder to estimate how well a neuronal population can predict a stimulus on a given trial (*Sanger, 1996*; *Jazayeri and Movshon, 2006*; *Graf et al., 2011*). We made two assumptions that allowed us to use this framework: (1) the firing rate can be described by Poisson statistics, and (2) firing rates across neurons are not correlated. Assuming that neuronal responses are statistically independent, the log likelihood of stimulus $s$ for the population response $r_i$ (where $i$ is equal to 1: $N$ total number of neurons) can be obtained by summing the log likelihoods of individual responses:

$$\log L(s) = \log\left[\prod_{i=1}^{N} p(r_i \mid s)\right] = \sum_{i=1}^{N} \log p(r_i \mid s)$$

where $f_i(s)$ is the speed tuning of neuron $i$.

Assuming that neuronal responses can be described by Poisson statistics, we compute the log likelihood of $s$ for the population response $r_i$ as following:

$$\log L(s) = \sum_{i=1}^{N} \log\left[\frac{f_i(s)^{r_i}}{r_i!} \exp(-f_i(s))\right] = \sum_{i=1}^{N} \log(f_i(s)) r_i - \sum_{i=1}^{N} f_i(s) - \sum_{i=1}^{N} \log(r_i!)$$

We can ignore the last term as it is stimulus $s$ independent. To avoid taking the *log(0)*, which is negative infinity, we assumed that at least one spike was fired for a given condition. The estimate corresponding to the population activity was then the speed and boundary position maximizing the log likelihood. This decoder has been widely used in previous studies (*Jazayeri and Movshon, 2006*; *Graf et al., 2011*; *Yanike and Ferrera, 2014*).

We performed cross-validation by using a leave-one-out approach (*Duda et al., 2001*). We trained the decoder on one randomly selected trial and tested the decoder on $n - 1$ trials. We avoided over fitting the data because the estimates of accuracy were not obtained on the same trials on which the decoder was trained. The neurons were recorded across days and varied in the number of trials for each speed and boundary position. To perform cross-validation we used the same number of trials ($n$ = 10) for each neuron in the population. If for a given speed there were more than $n$ = 10 trials, we randomly selected $n$ = 10 trials, otherwise, we reconstructed some of the trials (24% of trials for slow boundary position; 21% of trials for fast boundary position) for some cells by adding firing rate from a Poisson distribution with the mean equal to the actual data mean (26% and 27% percent of cells were reconstructed for slow and fast boundary position, respectively). To evaluate decoding accuracy (i.e., prediction accuracy, fraction of correct estimates) we randomly drew without replacement 1000 samples and then averaged the decoding accuracy for either speed or boundary position.

To evaluate the accuracy of the read out of population activity we only used trials when animals categorized stimuli correctly (i.e., correct), unless stated otherwise. The average prediction accuracy for each signal varied with subjects' performance, as it was significantly better on correct trials compared

to all trials for post-saccade (1-way ANOVA (trial type), p < 0.0001) and reward (p < 0.0001) populations. We also evaluated decoding accuracy as a function of the number of neurons (*Figure 7—figure supplement 1*). We obtained the average prediction accuracy (either across speeds or across boundary positions) for populations of different sizes N by randomly drawing without replacement N = [2 − max] separately for neuronal populations with activity during the post-saccade and reward periods of the task. We then normalized the average prediction accuracy by $\frac{\hat{\sigma} - chance}{1 - chance}$, where $\hat{\sigma}$ is the average prediction accuracy and chance level is 0.125 or 0.5 for speed or boundary position decoding.

To determine whether the prediction accuracy varied between trials when subjects categorized the same stimuli correctly and incorrectly, we only used a subset of stimuli, speeds 6 and 12, which had a sufficient number of incorrect trials. To obtained the prediction accuracy for speeds 6 and 12 during incorrect categorization, we randomly draw $n = 10$ trials from only incorrect trials using the same method as described above. We compared the prediction accuracy for speeds 6 and 12 between correct vs incorrect categorization near the boundary with correct categorization near vs far from the boundary. For these stimuli incorrect categorical judgments were followed by the failure to represent the identity of the preceding stimulus (*Figure 5A,C*), in contrast correct categorization of the same stimuli, either near or far from the boundary, were followed by a reliable but different read out of stimulus speed based on the population activity (2-way ANOVA [trial type, boundary], p < 0.001).

We also tested whether we could read out the direction of random dots motion (up or down) from neuronal population activity. The direction of dots motion was irrelevant feature of otherwise relevant stimuli, and animals were never asked to judge it. On a given trial, we calculated the prediction accuracy, as described above, with $s_j$ where $j = 1{:}32$ dimensions (where, 1:8, up, slow; 9:16, down, slow; 17:25, up, fast; 26:32, down, fast).

To obtain a representation of the boundary position in neuronal population activity, we used the log likelihood ratio to discriminate between pairs of neighboring speeds $s_1$ and $s_2$ (seven pairs, [2–4, 4–6, …, 14–16]). We obtained the log likelihood ratio as:

$$\log LR(s_1, s_2) = \log\left(\frac{L(s_1)}{L(s_2)}\right) = \log L(s_1) - \log L(s_2)$$

where, $s_1$ and $s_2$ are stimuli speeds on each trial. The decoder selected $s_1$ when $LR > 0$ and $s_2$ when $LR < 0$.

We used a permutation test for comparing $n$ samples, with Bonferroni correction for multiple comparisons at p values equal to $0.05/n$.

## Acknowledgements

We thank T Barnes, M Delgado, A Graf, B Lau, Y Naya, J O'Doherty, M Shapiro, and D Tsao for valuable comments and discussions. We also thank R Andersen for generous support and advice.

## Additional information

### Funding

| Funder | Grant reference number | Author |
| --- | --- | --- |
| National Institute of Mental Health | R01-059244 | Vincent P Ferrera |
| Robert E. Leet and Clara Guthrie Patterson Trust | Fellowship in Brain Circuitry | Marianna Yanike |
| Brain and Behavior Research Foundation | NARSAD Young Investigator Award | Marianna Yanike |

The funders had no role in study design, data collection and interpretation, or the decision to submit the work for publication.

### Author contributions

MY, Conception and design, Acquisition of data, Analysis and interpretation of data, Drafting or revising the article; VPF, Conception and design, Drafting or revising the article

## Ethics

Animal experimentation: All experimental procedures and treatments were in accordance with NIH guidelines and approved by the Institutional Animal Care and Use Committee at Columbia University (animal welfare assurance number A3007-01) and the New York State Psychiatric Institute (AWA number A4148-01). All surgery was performed under aseptic conditions using isoflurane anesthesia. The animals received postoperative analgesics during postsurgical recovery.

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
