## [Decision Letter]

Thank you for sending your work entitled “Interpretive monitoring in the caudate nucleus” for consideration at *eLife.* Your article has been favorably evaluated by Eve Marder (Senior editor) and 3 reviewers, one of whom is a member of our Board of Reviewing Editors.

The Reviewing editor and the other reviewers discussed their comments before we reached this decision, and the Reviewing editor has assembled the following comments to help you prepare a revised submission.

Reviewer# 1: His or her major concern relates to how authors dissociate the post-decision signals (post-saccade and reward responses) from a purely oculomotor response that simply reflects task difficulty. Can the author rule out this potential interpretation? Did the authors use a control task in which the monkeys simply made the same saccade responses in the absence of stimulus speed category? Authors should also be able to plot RTs and MTs as function of speed category with the two boundaries.

Reviewer # 2: While this reviewer thinks that the neuronal results are of interest and have potential, he or she believes that additional control analyses and discussion are needed to strengthen the results. In particular, additional consideration should be given to factors such as attention and arousal which are expected to covary with task difficulty, reward expectation and reward delivery (also in the same line as reviewer #1). This is important because you examine neuronal data in epochs following feedback about the animals' correct/error performance on each trial, and corresponding rewards. Furthermore, this reviewer and reviewer #1 question the quality of the writing, and suggest that care should be made to revise both the clarity of the prose and the associated grammar.

Reviewer # 3: This reviewer feels that the current results were not properly discussed in the context of the literature regarding the role of caudate in post-action evaluation for monitoring behavioral performance based in reward information. Are there any controls to rule out previous interpretations on the current database? Also, this reviewer would like more analysis on the presaccadic period with the same analytical tools used in your current version. He or she feels it critical to understand behavioral evaluation, by comparing the actual decisions with the outcomes. Although the BRE did not consider this very critical, this reviewer is concerned by the small database (38 neurons) used to characterize the functional properties of caudate in your task.

---

## [Author Response]

*Reviewer# 1: His or her major concern relates to how authors dissociate the post-decision signals (post-saccade and reward responses) from a purely oculomotor response that simply reflects task difficulty. Can the author rule out this potential interpretation? Did the authors use a control task in which the monkeys simply made the same saccade responses in the absence of stimulus speed category? Authors should also be able to plot RTs and MTs as function of speed category with the two boundaries*.

We thank the reviewer for raising this important point. We believe that the post-decision signals reported in our study are not simply oculomotor responses reflecting task difficulty. During the speed categorization task, the most difficult decisions occurred near the boundaries. Monkeys had a hard time categorizing the speed of stimuli 6 and 12 when they were near the boundary, but not when the same stimuli were away from the boundary. Our behavioral analyses showed that task difficulty was similar between these two stimuli when categorized near the boundary. We reasoned that if a neuron coded task difficulty then it would respond similarly to the most difficult stimuli, independently of the boundary position. In contrast to this hypothesis we found that most individual neurons were selective to a particular category/boundary. To illustrate this point, we show an example neuron with differential response to the same stimulus (speed 12) when animals classified it into different categories (orange/blue corresponds to ‘fast’/ ‘slow’ respectively; Figure 8 left panel). This neuron did not differentiate responses to any other stimuli, including speed 6, even though behaviorally it was also difficult to categorize. We calculated a category index for this neuron (the same index used in the paper to quantify whether neurons respond differently to the same stimuli depending on the boundary position) to show that it differentiated difficulty for only one boundary (Figure 8, right panel).Author response image 1.Neuronal activity and category index for an example neuron. Average neuronal activity plotted as a function of stimulus speed sorted by the boundary position (left). Category index for the same neuron showing maximum discrimination only near one category boundary.

In addition, as suggested by the reviewer, we did perform the control task in which monkeys made saccades to spatial targets without the speed stimulus. These control trials were intermixed with the categorization trials in 17 recording sessions. We found that while 7/17 neurons retained similar directional preference and response level, 10/17 neurons had a significant suppression in response during the control trials compared to the categorization trials. Thus, many caudate neurons were modulated by the task context and their oculomotor properties varied with the task at hand.

We also used a memory guided saccade task with 8 spatial targets at 45 degrees intervals to identify task-related neurons. Animals made saccades to spatial targets and were rewarded with either safe (p = 1.0) or risky (p = 0.5) reward outcome (50). As reported in the paper on page 15, we found that many neurons with strong direction selectivity only responded in the memory guided saccade task and not in the categorization task (21 out of 176 neurons). Also, none of the neurons with category-specific responses (24 neurons) in the categorization task had significant difference in response to risky and safe rewards in the memory guided saccade task. These results are consistent with the previous studies by [11], [12] showing that caudate neurons were strongly modulated by the task context during flexible decision making.

To address the last point raised by the reviewer we analyzed both RTs and MTs as a function of the speed category boundary. We found that monkeys’ RTs did not vary with the stimulus speed or boundary position, suggesting that subjects’ behavioral responses were automatic. Figure 9 shows the average RTs for each stimulus sorted by the boundary position for each monkey. We also tested whether there were differences in the MTs (not shown), but found no systematic variations with the speed of the moving dots. These results are not surprising because both animals were highly proficient in the categorization task, which requires flexibility but not deliberation or temporal integration of sensory evidence.Author response image 2.Reaction time vs. stimulus speed for each category boundary separately for each monkey.

*Reviewer # 2: While this reviewer thinks that the neuronal results are of interest and have potential, he or she believes that additional control analyses and discussion are needed to strengthen the results. In particular, additional consideration should be given to factors such as attention and arousal which are expected to covary with task difficulty, reward expectation and reward delivery (also in the same line as reviewer #1). This is important because you examine neuronal data in epochs following feedback about the animals' correct/error performance on each trial, and corresponding rewards. Furthermore, this reviewer and reviewer #1 question the quality of the writing, and suggest that care should be made to revise both the clarity of the prose and the associated grammar*.

We agree with the reviewer that whether or how attention and arousal modulate responses of caudate neurons is a very fascinating question. While we cannot directly address this question, because we did not manipulate attention/arousal in our task, we can outline different possibilities and provide tentative interpretations of the data. We tried to control for the effects of attention/arousal by having two category boundaries; since behavior is the most difficult around the boundary, it is equally difficult around each boundary. Reaction time is the most common behavioral index of arousal and attention. As discussed in the response to Review 1, RT did not vary as a function of stimulus proximity to the category boundary. Thus, a neuron coding attention/arousal would respond similarly between the two boundaries. Instead we found that many neurons differentiated responses near one, but not both boundaries. These results suggest that individual caudate neurons do not directly code attention/arousal during the flexible categorization task. However, the population level analysis revealed that cognitive information was more reliable near compared to away from the boundary. Thus, it is tempting to speculate that attention/arousal can possibly selectively enhance cognitive representation in the caudate nucleus to provide evaluation of specific on-going behavior (i.e. to improve read out of information on the most difficult decisions). Previously we found that the greater salience of risky outcome can possibly account for the better spatial selectivity in caudate neuronal population (50). Consistently, salience-related signals have also been found in the caudate nucleus (Asaad and Eskandar, 2011). These results suggest that the greater salience of uncertain or risky outcome can possibly account for better spatial and cognitive/contextual representation in the caudate nucleus.

Could the prospect of an uncertain outcome near the boundary modulate cognitive representation in the caudate? In other words, the reward outcome or reward probability also varied with the boundary. As the reviewer correctly pointed out, the monitoring signals reported here could be related to reward coding. To address this question, we asked whether the post-decision neuronal activity systematically varied with the probability of reward for each session. We found that right after the decision some neurons (9 out of 31, 29%) coded the expectation of reward by either decreasing (8 neurons; Figure 10) or increasing (1 neuron; Figure 10) their firing rate with the probability of reward. During the reward delivery, some neurons (12 out of 23, 52%) also coded the probability of reward by decreasing their firing rate (Figure 10). These results show that the monitoring signals in the caudate can be related to the coding of expectation of reward and/or uncertainty right after the decision and the probability of outcome during the reward delivery. We suggest that the signals related to reward shape categorical representation in the caudate. While these motivational signals were highly context specific at the level of individual neurons, their population read-out provided a global evaluative signal. These results suggest that the motivational signals are necessary for context specific decisions to adapt to rapidly changing conditions. We have modified the Discussion section to incorporate these comments.Author response image 3.Examples of individual neurons with neuronal activity during the psacc (A,B) and reward ( C ) periods of the task as a function of the probability of reward sorted by the boundary position (red/blue correspond to slow/fast boundary positions).

Also, we substantially revised our manuscript to improve the quality of writing.

*Reviewer # 3: This reviewer feels that the current results were not properly discussed in the context of the literature regarding the role of caudate in post-action evaluation for monitoring behavioral performance based in reward information. Are there any controls to rule out previous interpretations on the current database? Also, this reviewer would like more analysis on the presaccadic period with the same analytical tools used in your current version. He or she feels it critical to understand behavioral evaluation, by comparing the actual decisions with the outcomes. Although the BRE did not consider this very critical, this reviewer is concerned by the small database (38 neurons) used to characterize the functional properties of caudate in your task*.

That was a great suggestion. We have extended our Discussion on the post-action evaluation based on reward information. In addition, we have performed a number of analyses to see if the neurons tracked any reward information. We looked at whether they followed reward history across trials. We found no evidence for that. It is tempting to speculate that monkeys didn’t choose based on the reward value of each stimulus, but based on categorization uncertainty. By randomly selecting the category boundary on each trial, the outcome of previous trials is minimally useful. We’ve done analyses to show that there is no cost of boundary switching on either accuracy or reaction time (Figure 11); no evidence of a win-stay, lost-shift strategy (Figure 12); or effect of previous trial speed, boundary or outcome on the current trial (Figure 13).Author response image 4.No cost boundary switching.Author response image 5.No evidence for Win-Stay, Loose-Shift strategy.Author response image 6.No effect of previous trial on current choice.

Perhaps, the complexity of the task precluded animals from tracking the history of reward and instead their strategy was to solve each trial based on the current stimulus and boundary. We also analyze few sessions in which the reward amount was increased as the session progressed. Again, we found no change in the responses of neurons. However, a few cells did track the reward amount in a linear fashion, but they showed no modulated by the categorization task. We suggest that the neurons were more strongly modulated by the sensory cues and thus integrated all the information pertaining to the sensory cues. We would like to thank the reviewer for bringing up this important point.

We were also very curious about the pre-saccadic responses in the caudate in the context of the speed categorization task. However, we found that the activity of only a small fraction of neurons was modulated by the category boundary. Thus, we were not able to provide comparative analysis. One possibility is that the anterior caudate is primarily involved in post-decision monitoring when behavior is highly familiar yet flexible. This idea is consistent with recent findings by Ding and Gold, who have systematically compared neural correlates of perceptual categorization in three highly interconnected structures: FEF, LIP and caudate. They found that neurons with post-decision signals were more prevalent in the caudate compared to the other two areas (Ding and Gold, 2011).

We would like to comment about the reviewer’s concern about the number of neurons. While we focused our analyses on 38 neurons, the total number of trials was 19639. The task was very challenging and required collecting many trials to fully evaluate the rich flexible behavior. Thus we strived to get many trials per cell. We specifically designed our analysis to address this problem as we didn’t want to simply average the responses of these neurons, but to evaluate them as a population. Thus we believe that the type of analysis we chose compensates for the number of cells.